# SPATS1 (spermatogenesis-associated, serine-rich 1) is not essential for spermatogenesis and fertility in mouse

**Carlos A. Capoano[1], Luis Adrián Ortiz-Laquintana[1], Rosana Rodríguez-Casuriaga[1], Geraldine Schlapp[2], María Noel Meikle[2], Ana Paula Mulet[2], Martina Crispo[2], Ricardo Benavente[3], Adriana Geisinger[1,4]***

1 Department of Molecular Biology, Instituto de Investigaciones Biológicas Clemente Estable (IIBCE), Montevideo, Uruguay, 2 Transgenic and Experimental Animal Unit, Institut Pasteur de Montevideo, Montevideo, Uruguay, 3 Department of Cell and Developmental Biology, Biocenter, University of Würzburg, Würzburg, Germany, 4 Biochemistry-Molecular Biology, Facultad de Ciencias, Universidad de la República (UdelaR), Montevideo, Uruguay

* adriana.geisinger@gmail.com

**Data Availability Statement:** All relevant data are within the paper and its Supporting Information files.

## Abstract

SPATS1 (spermatogenesis-associated, serine-rich 1) is an evolutionarily conserved, testis-specific protein that is differentially expressed during rat male meiotic prophase. Some reports have suggested a link between *SPATS1* underexpression/mutation and human pathologies such as male infertility and testicular cancer. Given the absence of functional studies, we generated a *Spats1* loss-of-function mouse model using CRISPR/Cas9 technology. The phenotypic analysis showed no overt phenotype in *Spats1^-/-* mice, with both males and females being fertile. Flow cytometry and histological analyses did not show differences in the testicular content and histology between WT and knockout mice. Moreover, no significant differences in sperm concentration, motility, and morphology, were observed between WT and KO mice. These results were obtained both for young adults and for aged animals. Besides, although an involvement of SPATS1 in the Wnt signaling pathway has been suggested, we did not detect changes in the expression levels of typical Wnt pathway-target genes in mutant individuals. Thus, albeit *Spats1* alteration might be a risk factor for male testicular health, we hereby show that this gene is not individually essential for male fertility and spermatogenesis in mouse.

## Introduction

Mammalian spermatogenesis involves the execution of three successive gene expression programs: the mitotic proliferation of spermatogonia (i.e. meiotic precursor cells), meiotic divisions, and spermiogenesis (i.e. the differentiation process of round spermatids, which are the outcome of meiosis, into sperm) [1]. Due to the continuity and asynchronicity of the spermatogenic process [2], the three gene expression programs coexist within adult testes. Several studies have shown that the transcriptomes of meiotic and post-meiotic cells are extremely complex, and exhibit the expression of a very high number of specific genes (e.g. [3–5]).

**Funding:** This work was supported by Comisión Sectorial de Investigación Científica (CSIC), UDELAR, Uruguay (https://www.csic.edu.uy/), under an I+D Groups 2018 grant to AG and RB. CAC was awarded with a PhD scholarship from Agencia Nacional de Investigación e Innovación (ANII), Uruguay (https://www.anii.org.uy/). Experiments carried out at UATE, IPMontevideo, were supported by FOCEM (MERCOSUR Structural Convergence Fund; https://focem.mercosur.int/es/), COF 03/11. The funders had no role in study design, data collection and analysis, decision to publish, or preparation of the manuscript.

**Competing interests:** The authors have declared that no competing interests exist.

The alteration of spermatogenic gene expression programs is at the basis of numerous pathologies, including infertility [6–11] and testicular cancer [12]. However, owing to some intrinsic obstacles–namely the high cell heterogeneity of testicular tissue, and the lack of reliable *in vitro* cultures [13–16]–the studies on the molecular groundwork of spermatogenesis, including the identification of those genes that are essential for fertility, have been hampered. To assess the relevance of each of the reproductive genes/proteins, their functional characterization is required.

*Spats1* ("spermatogenesis-associated, serine-rich 1"; firstly called *Srsp1*) was previously identified in our laboratory as a testis-specific gene in rat [17]. The protein was first detected in embryo testes at 17.5–18 days post-coitum (dpc), where it was evident in gonocytes and peritubular myoid cells. Then it is accumulated during postnatal testis development reaching its maximum expression levels at 21 days postpartum (dpp) coincidentally with the pachytene stage, while signal intensity decreased in adult testes. Overall, the highest SPATS1 signal was observed in spermatocytes I (i.e. meiotic prophase), although relatively high protein levels were detected in Sertoli cells, spermatogonia, and myoid cells as well [18]. RNAseq studies showed that for mouse, the highest *Spats1* expression levels among different spermatogenic cell types were in primary spermatocytes as well [19]. We have also shown that SPATS1 is a highly phosphorylatable protein [18]. Moreover, SPATS1 is highly conserved along the evolution of metazoans, which allows to suspect that this protein could have an important, conserved role in testis [18].

More recently, a couple of reports have suggested a link between SPATS1 and male fertility. In one of these reports, the expression levels of *Spats1* were notably decreased in men with severely impaired spermatogenesis [20]. In the other—a genome-wide association study (GWAS)—*Spats1* was revealed as a potential candidate gene related to sperm quality in Holstein-Fresian bulls [21]. Furthermore, an exome-wide sequencing study has proposed a possible association between *Spats1* mutation and the development of human seminomas, which are the most common type of testicular cancers [22]. On the other hand, an existing report has suggested a link between SPATS1 and the canonical Wnt signaling pathway [23], a key regulatory pathway with roles in the control of diverse developmental processes and in cancer (e.g. [24–27]).

Nevertheless, no functional studies towards the elucidation of the role of *Spats1* in testis have been performed so far. As an attempt to contribute to the characterization of its function, we generated *Spats1* loss-of-function mice by means of CRISPR/Cas9 technology. Our results show that *Spats1* is not individually essential for mouse male fertility.

## Materials and methods

### Evolutionary conservation

For homology searches of mouse *Spats1*, we used the *Mus musculus Spats1* mRNA sequence from Genbank (NCBI Reference Sequence: NM_027649.3; this sequence corresponds to transcript ENSMUST00000024731.8 from Ensembl project database). For protein comparisons, the employed protein sequence was AAI31910 (A2RRY8 in the Uniprot database), of 269 amino acids. Sequence homology searches were made with the BLAST program (https://blast.ncbi.nlm.nih.gov/Blast.cgi), using blastp and tblastn tools. The comparisons were conducted alternatively against the entire database, against vertebrates exclusively, or excluding mammals (according to the parameters selected in each case). Sequence alignments were performed with ClustalW. A highly resolved tree for phylogenetic relationship between SPATS1 along the evolution of metazoans was built in MEGA (Molecular Evolutionary Genetics Analysis) [28].

For *Spats1* tissue-specific expression pattern analysis in different species, we collected all the available information in Expression Atlas (https://www.ebi.ac.uk/gxa/home) and NCBI databases.

## Animals

All animal procedures to generate the KO mice were performed at the SPF animal facility of the Transgenic and Experimental Animal Unit, Institut Pasteur de Montevideo (UATE, IPMontevideo). Experimental protocols were accordingly approved by the Ethics Committee for Animal Use ("Comisión de Ética en el Uso de Animales" [CEUA], protocol number 001–13), in accordance with national law 18,611 (Uruguay) and international animal care guidelines (Guide for the Care and Use of Laboratory Animals) (NRC, 1996). B6D2 F1 hybrid mice to generate the KO line were bred at the UATE.

All subsequent experimental animal procedures were performed at Instituto de Investigaciones Biológicas Clemente Estable (IIBCE, Montevideo, Uruguay), also in accordance with national law 18,611, and following the recommendations of the Uruguayan National Commission of Animal Experimentation ("Comisión Nacional de Experimentación Animal" [CNEA], approved experimental protocol 004/09/2011).

Male mice were humanely euthanized by cervical dislocation. Testes were measured and weighed individually, after removal of the tunica albuginea.

## Production of CRISPR/Cas9 knockout mice

Potential cleavage sites were retrieved with Custom gRNA software (Thermo Fisher Scientific, Waltham, MA, US), allowing for up to 3 mismatches in the single guide RNA (sgRNA) target sequence. The following sgRNA target, within the 3rd exon of *Spats1*, was chosen because it did not show any off-target sequences in the genome of *M. musculus*: TTCGCTGCCTGAAATCCCAA**AGG** (the protospacer adjacent motif [PAM] sequence is in bold characters). The sgRNA and Cas9 mRNA were ordered from Invitrogen (Carlsbad, CA, US). Validation of sgRNA/CRISPR cleavage activity was done by means of GenArt Genomic Cleavage Detection Kit (Invitrogen) following the instructions of the manufacturer, and using transfected NIH 3T3 cells.

Following validation of the sgRNAs, a mix of 50 ng/μL Cas9 mRNA and 20 ng/μL sgRNA diluted in microinjection buffer were microinjected into the cytoplasm of one-cell B6D2 F2 embryos. Approximately 200 zygotes per assay were microinjected, using a micromanipulator system (Transfermann NK2, Eppendorf, Germany). The microinjected zygotes were immediately transferred into the oviduct of recipient pseudopregnant females of the same strain (20–25 embryos/female). Full-term pups were naturally delivered 19–21 days later. Three backcrosses to C57BL/6J were done to obtain KO and WT littermates for the different experiments.

## Genotyping

At 21 days postpartum, immediately after weaning, the pups were appropriately numbered and approximately 0.5 cm of each tail tip was biopsied and used for genotyping. Genomic DNA was extracted from each tail tip with the GeneJET Genomic DNA Purification Kit (Thermo Fisher), and the region flanking the target site (465 bp) was amplified by standard PCR. The designed primers (*Spats1*-E3-Forward and *Spats1*-E3-Reverse) are listed in Supplementary S1 Table. PCR bands were extracted from agarose gel by means of the GeneJET Gel Extraction and DNA Cleanup Micro Kit (Thermo Fisher), and direct sequencing of the PCR products by Sanger sequencing from both ends was performed.

## Fertility test

*Spats1*[-/-] males (45 dpp) were caged with *Spats1*[-/-] females, in duplicate experiments for up to three months. In another experiment, *Spats1*[-/-] males (45 dpp) were caged each with 3 sexually mature WT females for the same time; six replicate experiments were performed. In parallel, *Spats1*[-/-] females were mated with 2 sexually mature WT males each for the same time lapse; four replicates were done in this case. The number of litters and pups from each pregnant female was recorded at birth. Experimental animals were compared with littermate controls when possible, or otherwise with age-matched non-littermate controls from the same colony.

For fertility tests of one-year old mice, *Spats1*[-/-] or WT males were mated with 3 WT females for three months in duplicate experiments.

## Western blot

Decapsulated testes were minced with a Teflon homogenizer and resuspended in Laemmli buffer. We loaded 15 μg of protein lysate per lane. Extracts were separated by SDS/PAGE (12% polyacrylamide), followed by liquid transfer to Protran nitrocellulose (Schleicher & Schuell, Germany). The membranes were blocked o.n. in TBST (Tris-buffered saline with 0.1% Tween-20) containing 10% nonfat milk, probed for 2 h with anti-SPATS1 primary antibody (sc-139435, 1:1,000; Santa Cruz Biotechnology, Dallas, TX, US) diluted in TBST, followed by 1 h incubation with anti-rabbit IgG HRP-linked secondary antibody (Pierce-Thermo Fisher, 1:30,000). Signal was detected by means of Supersignal West Pico Chemiluminescent Substrate Kit (Pierce-Thermo Fisher). Probing with anti-β-actin (ab8227, Abcam, Cambridge, UK) was used as a control for equal loading.

## Histological analysis

Whole testes were immersed in a fixative solution containing 2.5% glutaraldehyde in 0.1 M phosphate buffer (pH 6.9), and left overnight. Samples were cut into smaller pieces, washed with 0.1 M phosphate buffer (6 x 10 min), postfixed in 1% osmium tetroxide in 0.1 M phosphate buffer for 1 h, and washed again with the same buffer solution (3 x 10 min). Samples were dehydrated in ethanol and embedded in Epon (Durcupan, Fluka/Sigma-Aldrich, St Louis, MO, US) according to conventional procedures. Then, 250 nm sections were cut using a Power Tome XL ultra-microtome (Boeckeler Instruments, Tucson, AZ, US), stained with toluidine blue, examined by bright field microscopy under an Olympus FV300 microscope (Olympus, Tokyo, Japan), and photographed with an Olympus DP70 digital camera by means of DPController v. 1.1.1.65 software.

## Analysis of testicular cell populations by flow cytometry (FCM)

Testes from 45-day-old mice were dissected into 96 mm glass Petri dishes containing ice-cold separation medium (10% v/v FCS in Dulbecco's Modified Eagle's medium, DMEM, with high glucose and L-glutamine), and cut into 2–3 $mm^3$ pieces after removal of the tunica albuginea. Three to four of these pieces were immediately placed into a disposable disaggregator Medicon with 50 μm separator mesh (BD Biosciences, San Jose, CA, US) and processed in a Medimachine device (BD) as previously instructed [29, 30]. Afterwards, the cells were fixed in 70% ethanol o.n. at 4˚C, centrifuged for 5 min at 1,000 rpm, and the pellet was resuspended in 500 μL PBS. After 5 min at room temperature, propidium iodide solution (PI, Sigma-Aldrich; 1 mg/mL) and RNase were added to the cell suspension at a final concentration of 50 μg/mL each, at 0˚C in the dark, and staining was allowed to proceed for 10 min at 0˚C in the dark before analysis.

Cells were analyzed with a FACSVantage flow cytometer (BD) equipped with a water-cooled argon ion laser (Coherent, Innova 304) tuned to emit at 488 nm of excitation wavelength. Laser power was set to 100 mW and the fluorescence emitted from PI was collected in FL2 using a 575/26 band pass filter. A 70 μm nozzle was selected to perform FCM measurements. DNA QC particles (BD) were used to optimize fluorescence detection as well as to check instrument linearity and doublet discrimination performance. Analysis of the following parameters was carried out with CELLQuest software (BD): forward scatter (FSC-H), side scatter (SSC-H), pulse-area or total emitted fluorescence (FL2-A), and pulse-width or duration of fluorescence emission (FL2-W). Dot plots of FL2-A *vs* time were used as a control of fluorescence emission during sample analysis. Doublets were excluded from the analysis using dot plots of FL2-A *vs* FL2-W. The quantifications of the cell populations were also performed with CELLQuest. Half a testicle was used for each analysis, and 4 animals of each type were analysed in parallel.

## Sperm motility and morphology

For motility analysis, the cauda epididymis of either 50 dpp or 1 year old mice were cut, and sperm were allowed to swim out for 15 min in 500 μL of HTF modified medium (50% solution A [200mM NaCl, 0.2% glucose, 5 mM KCl, 0.7 mM $Na_2HPO_4.H_2O$, 1 mM $C_3H_3NaO_3$, 0.12 mg/mL penicillin G, 0.1 mg/mL streptomycin, 3.5 mM $CaCl_2.2H_2O$, 1 mM $MgCl_2$, 60% sodium lactate]; 50% solution B [50 mM NaHCO3]; 0.3% BSA). For each animal, one epididymis was incubated at 37˚C for 10 min in a $CO_2$ incubator before analysis, while the other was incubated for 60 minutes to promote in *vitro* capacitation. Five μL per slide were loaded, and observed under a Nikon E200 Eclipse microscope (Nikon Instruments, Tokyo, Japan). Sperm motility was assessed by computer-assisted sperm analysis (CASA). Movement characteristics were evaluated using an automated analyzer (SCA-Microoptics, Barcelona, Spain), with parameter settings for mouse. Semen from 3 KO and 3 WT individuals was evaluated in each case.

Sperm morphology was evaluated with an Olympus FV300 microscope under differential interference contrast (DIC) microscopy, using FluoView software (Olympus). For 50 dpp males we used 6 WT and 6 KO individuals, while for 1 year old males, 5 individuals of each type were employed. At least 200 sperm per individual were analyzed. Sperm heads were classified as morphologically normal or anomalous. In turn, anomalous heads were categorized as: without hook, banana-shaped, and "others", based on the categorizations of other authors [31, 32].

## qRT-PCR

Total RNA was extracted from WT and KO testes using PureLink RNA Mini Kit (Ambion-Thermo Fisher, Foster City, CA, US). RT and qPCR experiments were performed with the Power SYBR Green Cells-to-Ct Kit (Ambion-Thermo Fisher), in a qPCR CFX96 Touch™ Real-Time PCR Detection System (BioRad, Hercules, CA, US). For retrotranscription, 20 ng RNA were used in 50-μL final volume for 1 h at 42˚C in accordance with the instructions from the manufacturer, and 2 μL cDNA were used for each PCR reaction. All the reactions were performed with three biological replicas. A calibration curve was built with serial dilutions of *Ppp1cc* gene (protein phosphatase 1, catalytic subunit, gamma isozyme), which had been previously validated as a good housekeeping gene for mouse testis [19]. Absolute expression values of each gene in arbirtrary units were obtained according to the calibration curve defined with the *Ppp1cc* gene. Additionally, crossed validation was performed with *Tax1bp1* (human T-cell leukemia virus type I-binding protein 1), a second gene whose usefulness as a calibrator gene had been previously demonstrated as well [19]. The primer sequences are listed in S1 Table.

### Statistical analysis

For all statistical analyses, the results of at least three independent experiments were considered for each case. The differences between KO and WT mice were analyzed using one-way analysis of variance (ANOVA), and differences were considered as significant at p-values < 0.05.

## Results

### *Spats 1* is a highly conserved gene

An updated phylogenetic analysis expanded the conservation degree of SPATS1, revealing its presence in a high number of taxa, and showing that *Spats1* originated early in evolution. In particular, blast analyses identified SPATS1 in 202 animal species including cnidarians, mollusks, brachiopods, hemichordates, echinoderms, urochordates, cephalochordates, cartilaginous and bony fish, amphibians, reptiles, and mammals including Monotremes, Metatherians, and all Eutherian orders (Fig 1A and S1 Fig). Curiously, no homologies were found in birds, with the only exception of the order Apterygiformes (i.e. the kiwi from New Zealand), where a predicted protein with a certain homology degree to mouse SPATS1 was identified. Similarly, we have not found homologs in members of the Ecdyzoa clade, including *D. melanogaster* and *C. elegans*. The absence of homologies in this clade has been previously reported for other mammalian proteins comprising meiotic proteins such as for instance synaptonemal complex components [33], and has been interpreted to be a consequence of the rapid evolution and sequence diversification of ecdyzoans [33, 34]. No significant similarities were found in plants either. Thus, despite the few above-mentioned exceptions, SPATS1 is highly conserved along the evolution of metazoans. Particularly, the extensive similarity between mouse and human SPATS1 can be seen in Fig 1B (see also S2 Fig for similarities at the nucleotide level).'

In order to evaluate if the expression pattern of *Spats1* was also conserved between species, we performed data mining. Expression atlas showed that *Spats1* is differentially expressed in the testes compared to other tissues, and this is true for the different analyzed mammalian species, including rat, mouse, man, macaque, horse, bull, sheep, and the marsupial opossum (not shown). Besides, for *Spats1* homologues identified beyond mammals and with available information at the NCBI database, a high proportion corresponded to mRNAs isolated from testis. Examples of this are the *Spats1* homolog from bony fish *Oryzias latipes* (GenBank Access number FS557457), cartilaginous fish *Callorhinchus mili* (access number XP_007901502.1), cephalochordates *Branchiostoma belcheri* and *Branchiostoma floridae* (access numbers XP_019647808 and XP_002590276.1, respectively), hemichordate *Saccoglossus kowalevskii* (access number XP_002731024.1), echinoderm *Strongylocentrotus purpuratus* (access number XP_001184069.1), brachiopod *Lingula anatina* (access number XP_013389114.1), and the mollusk *Octopus bimaculoides* (access number XP_014774497.1), among others. Of note, a recent study in juvenile snakeskin gourami fish also identified *Spats1* among the top differentially expressed genes in testis, as compared to ovary [35].

Thus, both the *Spats1* gene and its testicular expression pattern appear to be conserved along the evolution of metazoans.

### *Spats1*⁻/⁻ mice are fertile and present normal spermatogenesis

The high evolutionary conservation, together with its testis-biased expression pattern, suggested that *Spats1* could be an important gene for male reproduction. To investigate this, we generated *Spats1*⁻/⁻ mice by obtaining a 2-bp deletion in exon 3 of the gene, with the use of CRISPR/Cas9 system (Fig 2A and 2B). The deletion resulted in a frameshift at residue 90, to

# A

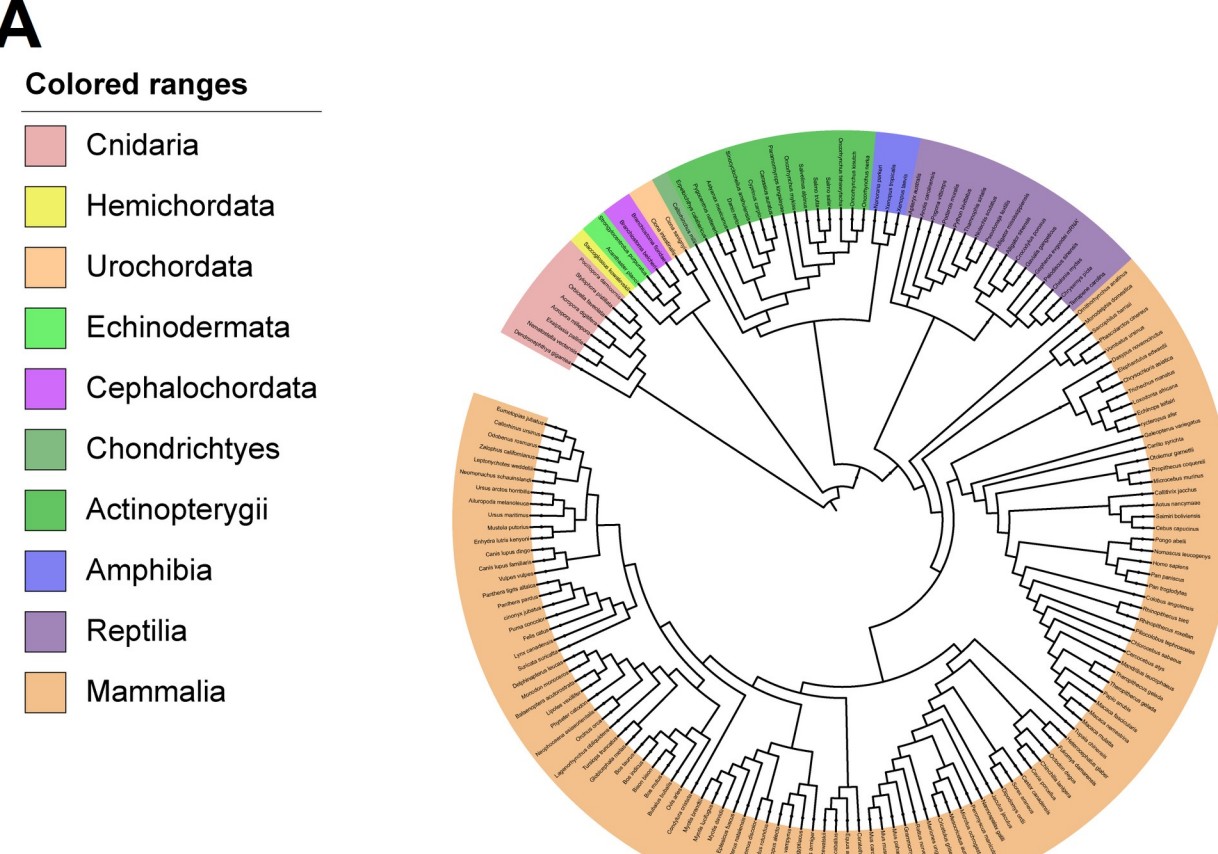

# B

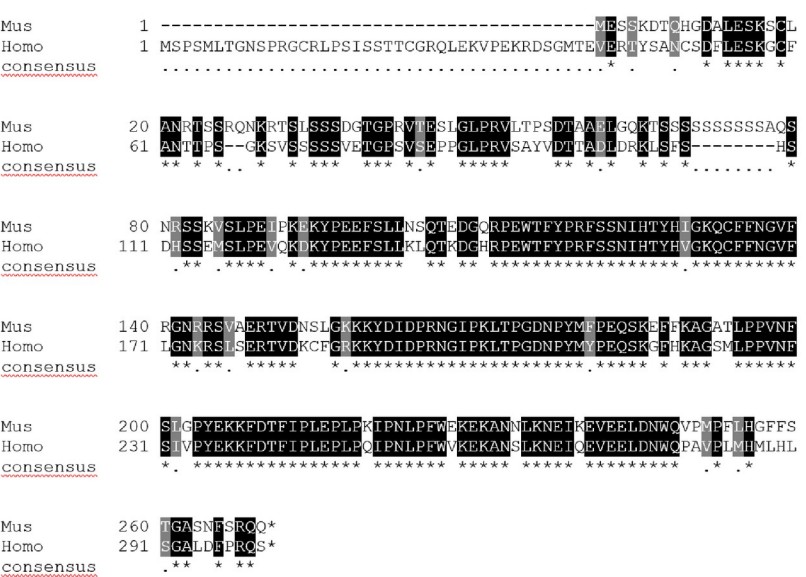

**Fig 1. Conservation of SPATS1 in metazoans.** (A) Highly resolved tree showing the presence of SPATS1 and its phylogenetic relationship across different metazoans taxa. The tree is drawn to scale, with branch lengths measured in the number of substitutions per site. (B) Alignment of

SPATS1 amino acidic sequence from mouse (*Mus musculus*) and man (*Homo sapiens*). Alignment was performed with ClustalW and BoxShade. Identical amino acids are highlighted in black and similar amino acids in grey.

generate a putative aberrant protein of 111 amino acids instead of the WT, 269-residue protein. Western-blot analysis with an antibody raised against the central region of SPATS1 detected the protein in WT animals, but not in *Spats1*-/- individuals (Fig 2C).'

The analysis of mutant mice showed no overt defects in *Spats1*-/- mice, either male or female. Homozygous mutants of both genders were fertile, and produced offspring of a litter size similar to WT ones (Fig 3A). Besides, no differences in testicular morphology, size or weight, were observed between WT and KO mice (Fig 3B).'

In order to have quantitative comparative analyses, testicular cell suspensions from adult WT and *Spats1*-/- mice were stained with propidium iodide and analyzed by flow cytometry (FCM), mainly based on DNA content [36]. No significant differences between WT and mutant mice were found, indicating similar relative proportions of either somatic cells and spermatogonia (2C in DNA content), meiotic cells (4C), spermatids (C) (Fig 3C), and spermatozoa (C, apparently sub-haploid left-most peak) (S2 Table) [37].

Histological analysis of young adult mice testes revealed no obvious differences between those of WT and *Spats1*-/- animals (Fig 3D). Furthermore, no significant differences in motility per cauda epididymis either for fresh or for capacitated sperm (Fig 3E and 3F), or in epididymal sperm concentration (S3 Fig), were observed among WT and *Spats1*-/- mice. Concerning sperm morphology, no significant differences in the proportion of anomalous spermatozoa in KO mice compared to WT ones were observed either (Fig 3G).

It has been reported that in some cases of KO mice no testicular phenotype is observed in young animals, but as the age advances a phenotype starts to develop, such as progressive fertility loss, decreased sperm count [38], or germ cell loss [39]. Therefore, we let some of the

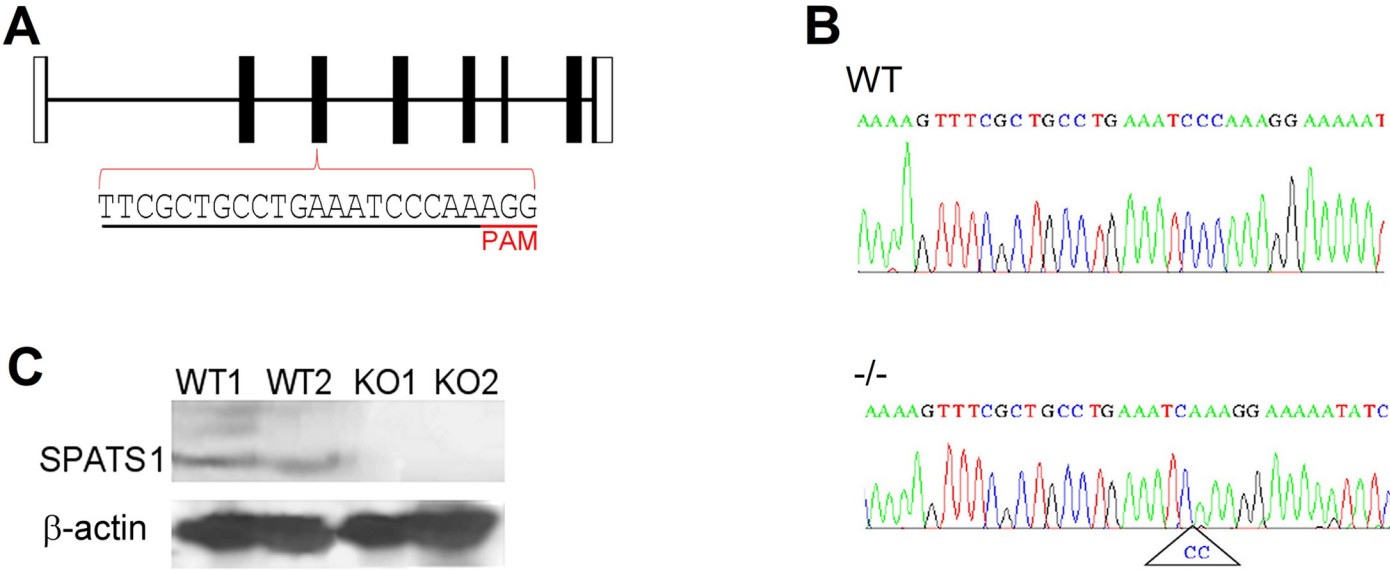

**Fig 2. Genomic structure and knockout strategy of *Spats1*.** (A) Graphical representation of the structure of mouse *Spats1*. Coding exons appear as solid black bars and non-coding exons as white bars. Introns are represented as horizontal lines. Sequence of the sgRNA, designed to target exon 3, is underlined in black, and the PAM sequence is underlined in red. (B) Representative genotyping results obtained through sequencing of PCR products amplified from mouse tail tips. The 2 bp deletion in *Spats1*-/- mutants is shown. (C) Western blot analysis showing the detection of SPATS1 in testicular lysates of two WT mice, but not in those of 2 KO ones. β-actin was used as a loading control.

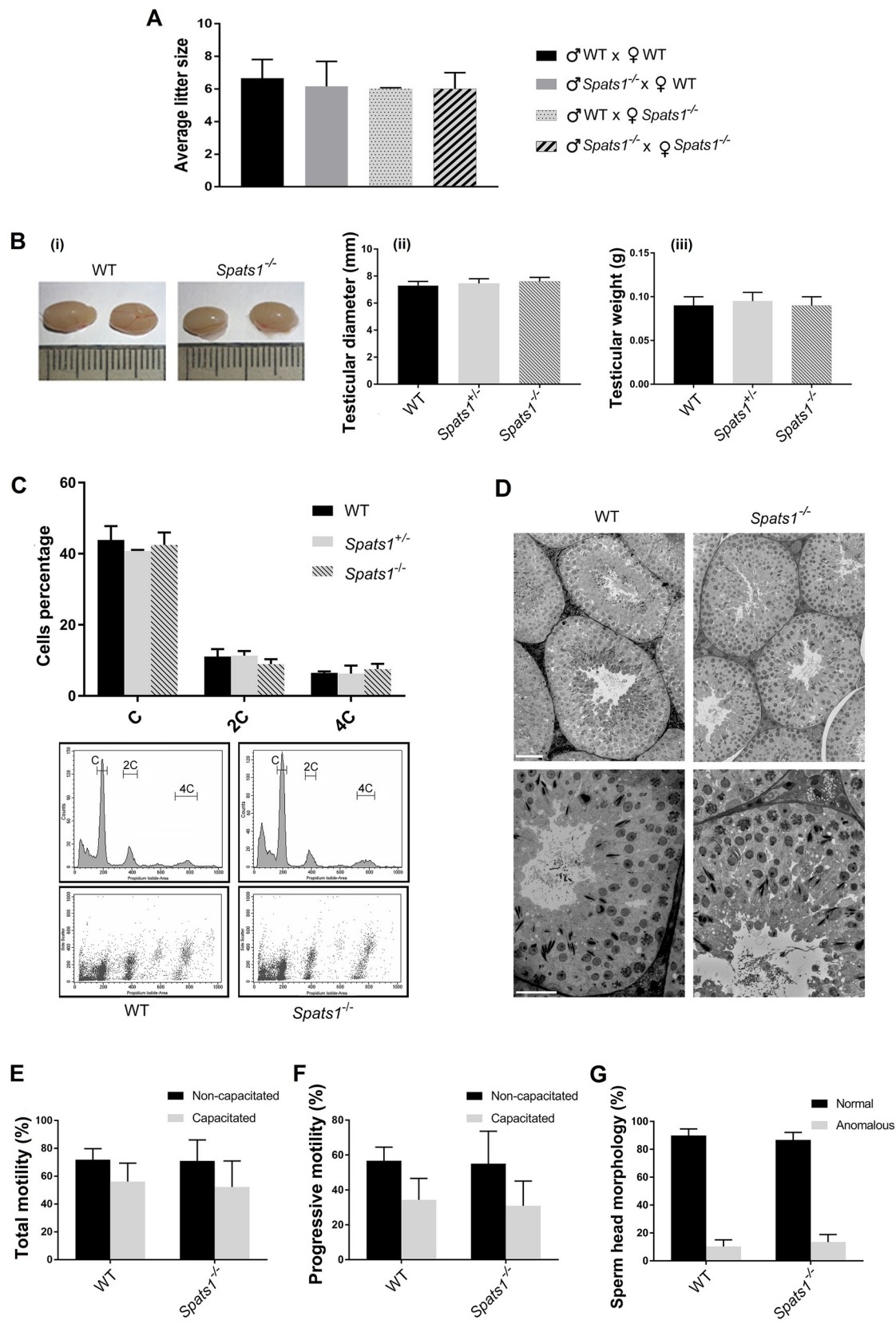

**Fig 3. Fertility and spermatogenesis analyses of *Spats1*<sup>-/-</sup> young adult mice of 45–60 dpp.** (A) Number of offspring per litter, obtained by intercrossing WT individuals, crossing WT females with KO males, WT males with KO females, and KO males and females. (B) Representative images of testes from WT and *Spats1*<sup>-/-</sup> male mice (i), and comparison of testicular diameter (ii) and weight (iii) of WT, *Spats1*<sup>+/-</sup>, and *Spats1*<sup>-/-</sup> animals. (C) Flow cytometric (FCM) analysis of testicular cell suspensions from WT, *Spats1*<sup>+/-</sup>, and *Spats1*<sup>-/-</sup> mice. Representative FCM profiles (histograms and dot plots) from WT and *Spats1*<sup>-/-</sup> are shown below. (D) Histological analyses of the testes of WT and *Spats1*<sup>-/-</sup> mice. Scale bar in the upper images corresponds to 50 μm, while scale bar in the lower images corresponds to 25 μm. (E) Total motility (percentage) of sperm from cauda epididymis of WT and KO mice. Non-capacitated and capacitated sperm were analyzed by CASA. (F) Percentage of sperm from cauda epididymis of WT and KO mice with progressive motility, as assessed by CASA. Again, non-capacitated and capacitated sperm are shown. (G) Percentage of sperm from cauda epididymis with normal head morphology, from WT and KO mice. All the data correspond to the analysis of 3–6 animals of each type, and are presented as the means ± SD.

animals reach 1 year of age and analyzed their phenotype. *Spats1*<sup>-/-</sup> mice at 1 year of age did not present any evident morphological alteration in comparison to WT, and the males presented testes of normal size and aspect (not shown). Fertility tests showed a slight although not significant difference with their WT littermates (Fig 4A). Histological analysis showed seminiferous tubules with similar appearance to those of WT animals (Fig 4B). Moreover, no significant differences in sperm motility (Fig 4C and 4D) and morphology (Fig 4E) between WT and KO mice were observed. Besides, although some differences in sperm concentration were observed in this case (S3 Fig), again, they were not significant.'

## No differences in the expression level of genes from the Wnt pathway are detected between the testes of *Spats1*<sup>-/-</sup> and WT mice

An existant report by Zhang *et al.* [23] suggested that SPATS1 would be a negative regulator of the canonical Wnt pathway. Using transfection of diverse molecular constructions into heterologous somatic cells, in that work it was observed that SPATS1 (therein renamed as DDIP [Dishevelled-DEP domain interacting protein]) was capable of binding and sequestering DVL2 (Dishevelled 2), a positive regulator of the Wnt pathway. According to their results, SPATS1 would weaken the β-catenin/TCF4 interaction by promoting the degradation of transcription factor TCF4 [23]. As spermatogenesis requires a fine-tuning of the Wnt pathway (e.g. [40–42]), we wondered if this pathway would be disregulated in the testes of *Spats1*<sup>-/-</sup> mice. In order to answer this question, we analyzed the expression levels of some of the most common Wnt pathway-target genes that are activated by the β-catenin/TCF4 complex, namely *Lef1*, *Tcf1*, Cyclin D1 (*Ccnd1*), and *c-Myc* [39, 41, 43–45], in the testes of *Spats1*<sup>-/-</sup> mice, in comparison to WT ones.

The analysis by qRT-PCR showed no significant differences in the expression levels of the target genes of the Wnt pathway in testis between WT and KO mice (except for *Lef1*, which contrary to what expected, showed lower expression levels in KO mice; Fig 5A). Particularly, *c-Myc* (for which a role in immature Sertoli cells has been suggested, and whose mRNA is usually not detectable in adult testes [46]) did not show expression above background either in the testes of WT animals or in those of *Spats1*<sup>-/-</sup> ones. Likewise, we did not detect expression of *Dvl2* in the testes of either WT or mutant animals. We also analyzed the expression of *Dvl1*, a *Dvl2*-related family member with probable functional redundancy [47] that shows stage-dependent expression changes along mouse spermatogenesis [48]. No significant differences in the expression levels of *Dvl1* were observed either.'

*Wnt4*, one of the initiators of Wnt signaling, is a key gene for the development of female genitalia, with its absence leading to partial female-to-male sex reversal in mouse (e.g. [42, 49]). A fine tuning of *Wnt4* expression is also implicated in testis development and spermatogenesis [50–55]. Interestingly, in a study *Spats1* appeared within the list of upregulated genes in *Wnt4*-null fetal mouse ovaries, compared to controls [56]. Therefore, here we also evaluated

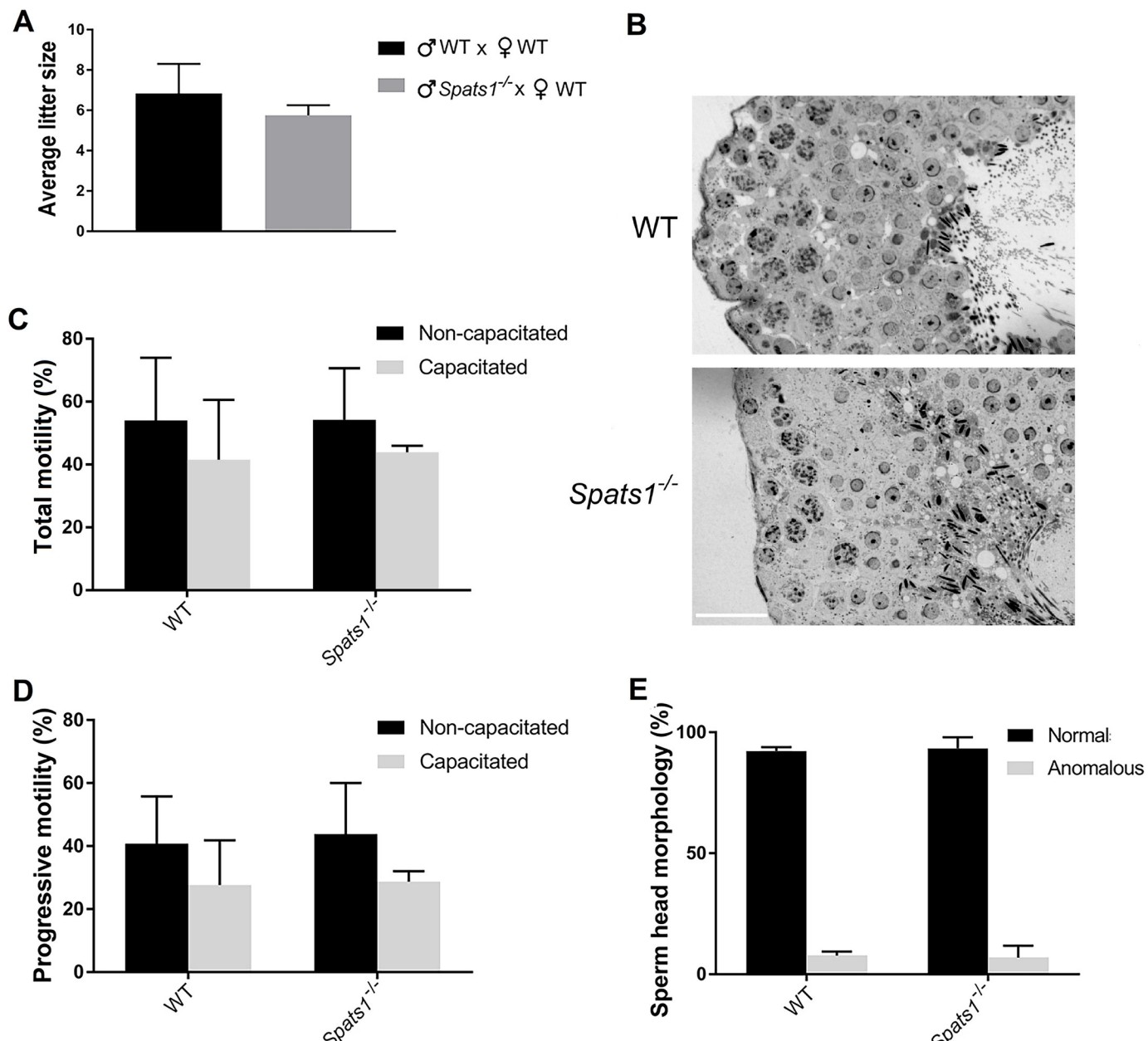

**Fig 4. Fertility and spermatogenesis analyses of *Spats1*<sup>-/-</sup> 1 year old mice.** (A) Average litter size, obtained by intercrossing WT individuals, or crossing KO males with WT females. (B) Histological analyses of the testes of WT and *Spats1*<sup>−/−</sup> male mice. Scale bar: 25 μm. (C) Total motility (percentage) of sperm from cauda epididymis of WT and *Spats1*<sup>−/−</sup> mice, as assessed by CASA. Non-capacitated and capacitated sperm were analyzed. (D) Percentage of sperm with progressive motility, from the cauda epididymis of WT and KO mice. (E) Percentage of sperm from the cauda epididymis with normal head morphology, from WT and KO mice. All the data correspond to the results obtained from the analysis of 3–6 individuals of each type, and are presented as the means ± SD.

the expression of *Wnt4*; its mRNA is usually undetectable in normal testes [51], and remained the same in those of KO mice. No differences in the expression levels of the gene coding for the core effector of the pathway, β-catenin (e.g. [25, 40, 42, 57]), were observed either (see Fig 5A).

Progressive spermatogenic defects have been reported for mutant mice with germ cell-specific constitutive activation of β-catenin as age increases [39]. As a consequence, we also

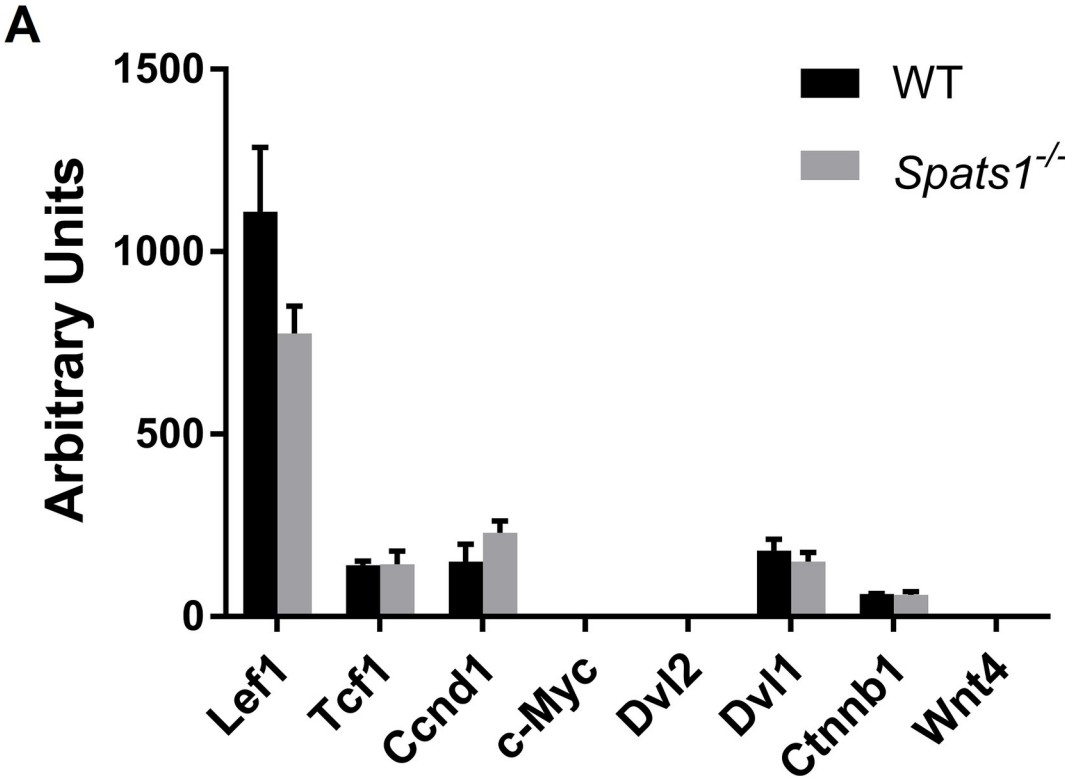

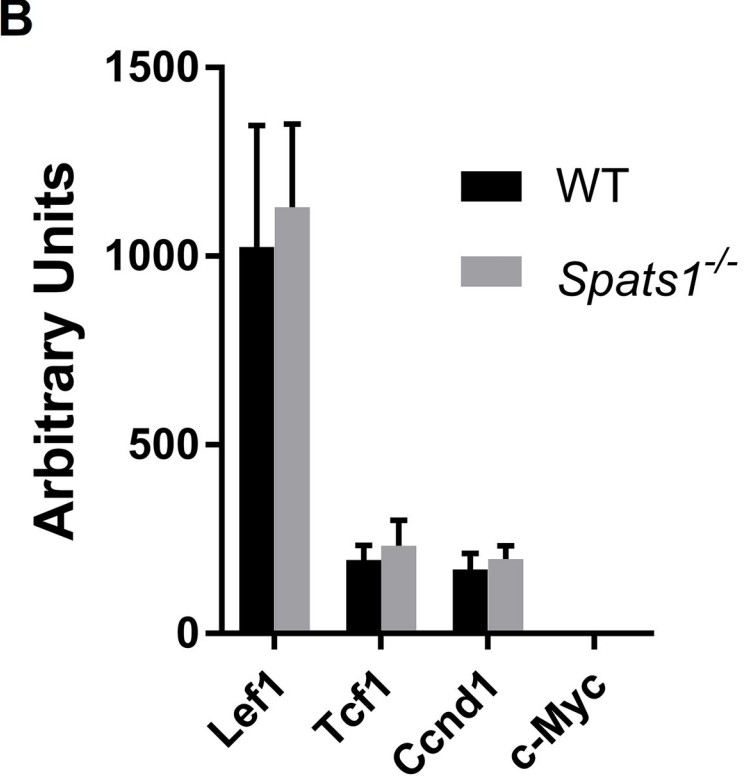

**Fig 5. Comparative expression analysis of genes of the Wnt pathway in *Spats1⁻/⁻* and WT mice.** (A) qRT-PCR of testicular RNA from 55 dpp WT and KO mice. (B) qRT-PCR of testicular RNA from 1 year-old WT and KO mice. Data are expressed as

absolute normalized expression levels in arbitrary units (mean ±SD). *Lef1*: *Lymphoid enhancer binding factor 1; Tcf1*: *Transcription factor 1; Ccnd1*: *Cyclin D1; c-Myc*: *Myc proto-oncogene; Dvl2*: *Dishevelled-2; Dvl1*: *Dishevelled-1; Ctnnb1*: *β-catenin; Wnt4*: *Wingless family member 4.*

analyzed the expression of the above-mentioned downstream targets of the Wnt pathway in the testes of 1 year-old KO mice. However, no significant differences between WT and *Spats1*[−/−] animals were observed ([Fig 5B]).

## Discussion

Some data have allowed us to suspect the importance of SPATS1 for testis development and/or spermatogenesis. First, it shows a testis-restricted expression pattern, and high expression levels in pachytene spermatocytes during the first spermatogenic wave. Second, it has a high evolutionary conservation. As reproduction-related proteins are in general poorly conserved–most probably due to unusually fast evolution rates [58]–this high conservation would suggest an important role for SPATS1. Moreover, a few reports have proposed its link with spermatogenesis, male fertility [20, 21], and testicular cancer [22]. Therefore, and due to the absence of functional studies, we addressed the generation of a *Spats1* KO mouse model for the analysis of its functional roles in relation to mouse spermatogenesis and fertility.

As expected for the KO of a testis-specific gene, *Spats1*[−/−] mice were viable, and female individuals were completely normal and fertile. However, unexpectedly, mutant males were fertile as well, and produced offspring of a similar litter size to WT ones. Besides, they showed no differences neither in testicular morphology, size or weight, nor in testicular cell content or histology. In addition, no significant differences in sperm count, morphology or motility between WT and KO mice were observed.

This work adds to an increasing number of studies that show a lack of phenotype for KO mice of testis-specific genes. As an example, Miyata et al. [59] reported the study of KO mice for 54 evolutionarily conserved genes with testis-enriched expression, including some genes believed to be important for fertility. They found that all the KO mutants were fertile, which indicates that those genes were not individually essential for male fertility. A similar result was obtained in another study where KO mice for 30 conserved testis-enriched genes were generated [60]. A possible explanation for the lack of phenotype may be functional redundancy [59, 60]: as reproduction is the "master function", it is likely that redundant genes to cover a same essential role exist, to ensure that it can be satisfactorily fulfilled. This could be the case for some reproductive genes where related family members may compensate for the KO gene-loss of function (e.g. [61–64]). Although SPATS1 would not be a member of a protein family, we cannot rule out the possibility of other proteins with overlapping function.

Sometimes reproduction-linked phenotypes of KO mice may become evident with age [38, 39]. However, this does not seem to be the case for *Spats1*[−/−] animals, as we failed to detect differences with WT ones in 1 year-old individuals.

On the other hand, it must be recalled that reproduction is extremely sensitive to the environment, and therefore some reproduction-related genes may not be essential in normal laboratory conditions but be required under other, more stressful circumstances. Moreover, disruption of a single gene may not cause an evident effect on fertility, but it may have dramatic consequences in combination with environmental factors, or with other mutations or polymorphisms [8, 65]. If this were the case for *Spats1*, it could possibly reconcile the discrepancies between the lack of phenotype of *Spats1* KO mice, and the reports that associate alterations or polymorphisms of *Spats1* with human male pathologies such as infertility and

testicular cancer [20, 22]. Furthermore, we cannot exclude the possibility that the absence of SPATS1 be better tolerated in mice than in humans.

Finally, a report using transfections in heterologous cell culture suggested that SPATS1 would be a negative regulator of the Wnt pathway [23]. We considered this an attractive idea, as accumulating evidence indicates the relevance of Wnt signaling in testis development and physiology, and its alteration in different male reproductive anomalies (e.g. [40, 42, 66]). However, when comparing *Spats1* KO *vs* WT mice, we could not find any significant change in the expression of common Wnt pathway-target genes in the testis, which is the tissue where *Spats1* is mostly expressed. In this regard, we remark the relative validity of heterologous somatic cell cultures and transpolation of results to the complex testicular tissue, and the obvious importance of performing *in vivo* studies [67], such as the generation of loss-of-function mutants. Thus, although of course we cannot rule out the possibility of a relation between SPATS1 and the Wnt pathway, we must state that in this study we did not find any evidence of this relation in testis.

In summary, this study adds *Spats1* to the list of evolutionarily conserved genes with a testis-restricted/differential expression pattern that are individually dispensable for mouse spermatogenesis and fertility. While according to other reports a defect in *Spats1* may be a risk factor for male testicular health in human and bull [20–22], we here show that the lack of SPATS1 protein alone does not cause an evident phenotype in mouse. This information is important for the scientific community in order to avoid duplicate efforts from other laboratories. Besides, it is useful in the search for potential male contraception targets, as it discards *Spats1* as a potential target.

## Supporting information

**S1 Table. Primers used in this study.**
(PDF)

**S2 Table. Quantification of the different cell populations in testicular cell suspensions from WT, *Spats1*[+/-] and *Spats1*[-/-] mice, by FCM.**
(PDF)

**S1 Fig. BLAST alignment of SPATS1 homologous sequences from different metazoan species.** *Mus musculus* protein sequence was used as query. Alignment was performed with ClustalW, and visualized using Jalview 2.11.1.4 (https://mybiosoftware.com/jalview-2-6-1-multiple-alignment-editor.html). The most conserved positions along evolution (from 0 to 9) are shown below.
(TIF)

**S2 Fig. Alignment of *Spats1* coding sequences from mouse (*Mus musculus*) and man (*Homo sapiens*).** Alignment was performed with ClustalW and BoxShade.
(TIF)

**S3 Fig. Relative sperm count (millions/mL) from cauda epididymis of WT and KO mice.**
(A) 45–60 dpp animals. (B) One year old individuals. Although the count is not completely reliable in absolute numbers (as obtained through swimming out from the cauda), the data is comparable between samples.
(TIF)

**S1 Raw images.**
(PDF)

## Acknowledgments

We thank BSc Gabriel Fernández-Graña (UATE, IPMontevideo) for animal care and cross-breeding. We are grateful to Dr Rossana Sapiro (Histology Department, Faculty of Medicine, UdelaR) for generously allowing us to use the CASA equipment, and for her help with it. MSc Federico Santiñaque is also thanked for his excellent technical assistance with FCM.

## Author Contributions

**Conceptualization:** Adriana Geisinger.

**Formal analysis:** Carlos A. Capoano, Luis Adrián Ortiz-Laquintana, Adriana Geisinger.

**Funding acquisition:** Ricardo Benavente, Adriana Geisinger.

**Investigation:** Carlos A. Capoano, Luis Adrián Ortiz-Laquintana.

**Methodology:** Carlos A. Capoano, Geraldine Schlapp, María Noel Meikle, Ana Paula Mulet, Martina Crispo.

**Project administration:** Adriana Geisinger.

**Resources:** Geraldine Schlapp, María Noel Meikle, Ana Paula Mulet, Martina Crispo.

**Supervision:** Rosana Rodríguez-Casuriaga, Ricardo Benavente, Adriana Geisinger.

**Visualization:** Carlos A. Capoano, Luis Adrián Ortiz-Laquintana.

**Writing – original draft:** Adriana Geisinger.

**Writing – review & editing:** Rosana Rodríguez-Casuriaga, Martina Crispo, Ricardo Benavente, Adriana Geisinger.

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
