## [Decision Letter · Decision Letter 0]

15 Mar 2021

PONE-D-21-03796

SPATS1 (Spermatogenesis-associated, serine-rich 1) is not essential for spermatogenesis and fertility in mouse

PLOS ONE

Dear Dr. Geisinger,

Thank you for submitting your manuscript to PLOS ONE. After careful consideration, we feel that it has merit but does not fully meet PLOS ONE’s publication criteria as it currently stands. Therefore, we invite you to submit a revised version of the manuscript that addresses the points raised during the review process.

Please include additional information on the functional role of SPATS in relation to evolutionary process. The functional relevance of the genes analysed of a particular pathway be elaborated in the discussion. Reanalyze the sperm count with tissue homogenates and using stage specific sections for histological anlyses are recommended.

We look forward to receiving your revised manuscript.

Kind regards,

Suresh Yenugu

Academic Editor

PLOS ONE

Journal Requirements:

3. Thank you for including your ethics statement:  "All animal procedures to generate the KO mice were performed at the SPF animal facility of the Transgenic and Experimental Animal Unit, Institut Pasteur de Montevideo (UATE, IPMontevideo). Experimental protocols were accordingly approved by the institutional Animal Ethics Committee (protocol number 001-13), in accordance with national law 18,611 (Uruguay) and international animal care guidelines (Guide for the Care and Use of Laboratory Animals) (NRC, 1996). B6D2 F1 hybrid mice to generate the KO line were bred at the UATE.

All subsequent experimental animal procedures were performed at Instituto de Investigaciones Biológicas Clemente Estable (IIBCE, Montevideo, Uruguay), also in accordance with national law 18,611, and following the recommendations of the Uruguayan National Commission of Animal Experimentation (CNEA, approved experimental protocol 004/09/2011).

Male mice were humanely euthanized by cervical dislocation. ".

Please amend your current ethics statement to include the full name of the ethics committee that approved your specific study.

For additional information about PLOS ONE submissions requirements for ethics oversight of animal work, please refer to http://journals.plos.org/plosone/s/submission-guidelines#loc-animal-research  

Reviewers' comments:

Reviewer's Responses to Questions

**Comments to the Author**

1. Is the manuscript technically sound, and do the data support the conclusions?

Reviewer #1: Partly

Reviewer #2: Yes

2. Has the statistical analysis been performed appropriately and rigorously? 

Reviewer #1: Yes

Reviewer #2: Yes

3. Have the authors made all data underlying the findings in their manuscript fully available?

Reviewer #1: Yes

Reviewer #2: No

4. Is the manuscript presented in an intelligible fashion and written in standard English?

Reviewer #1: Yes

Reviewer #2: Yes

5. Review Comments to the Author

Reviewer #1: In this manuscript, Capoano and colleagues report the generation of a KO mouse model for the gene SPATS1. This Serine-rich gene is the member of a 3 gene-family (including SPATS1, 2 and L2) and is found in many animals. Its expression has been reported in the testis either prior or during meiosis.

The authors used CRISPR technology to induce a frame-shifting mutation in the 3rd exon of SPATS1. The manuscript investigates the potential defects in reproductive fitness of KO animals and finds no major alteration in germ cell development or mating success. Thus, the author conclude that it is not essential for fertility in mouse.

The work is original, as no other SPATS1 KO mouse models have been published to my knowledge. The research is presented clearly and appropriately. Overall, this manuscript meets all the standard of research ethics and conclusions are supported by the data - with the few clarifications needed and described below.

In the following section, I provide revision points which should clarify and strengthen the manuscript.

1) In the abstract (i.e. last sentence) and throughout the text the authors should be careful about making parallels between Human fertility phenotypes related to SPATS1 and the mouse phenotypes described here. It is very possible – as the authors briefly note in the discussion – that one is not a good model for the other, as it is the case with many other spermatogenesis genes.

The gene is present, that seems clear. Yet, considering the scope of the study, I would recommend showing an alignment of mouse and human SPATS1 sequences (CDS and Proteins) to point to the regions that are indeed “conserved”.

Please also carefully consider the next comments regarding SPATS1 evolution in the following points.

2) Page 4 Line 78-80: What is the proposed function across evolution? Meiosis? Is it a sex-specific gene other metazoan?

3) Page 12 Line 280-281: Clarify this sentence. What is the observation?

4) The conclusion from figure 1 need to be developed to fully support the authors conclusions: the authors need to show an alignment or a few representative sequences to support the claim that it is “highly conserved” (again to distinguish between evolutionarily retained and conserved at the sequence level). Alternatively, they could use branch length on the tree to support this claim. Finally, the authors need to specify what type of tree is displayed and the scale used (substitutions per site maybe?) in Fig1.

5) Page 13 line 299: What does differentially expressed means here? Relative to what? Please clarify.

6) Can the author specify how many backcrosses in which strain background was made to produce the experimental set of mice?

7) Can the author specify where the antibody maps relative to the full-length protein and frame-shifted (truncated) sequence?

8) Page 17 Line 412-415: maybe the author could show results of these qPCRs for 4C cells.

9) Page 17 Line 419: please indicate here what is the expected level and function of c-Myc in WT testes.

10) Page 21 Line 483: I would not be as categorical that SPATS1 is *not* a member of a multi-gene family. There is also SPATS2 and 2L in the mouse and human genome. Not to mention the SPATA gene family…

Reviewer #2: Thank you for the opportunity to review this article. The authors generated a knockout Spats1 mouse model to investigate the role of SPATS1 in male fertility. Knockout male and female mice were fertile and showed no signs of any reproductive defects. While the paper is mostly rounded, I provide a few comments below and suggestions that I believe will improve the robustness of the article. I recommend that the article is accepted conditionally to these being completed.

Comments:

Line 32 – SPATS1 loss of function or overexpression? – I note this is further fleshed out well below in lines 81-91

Line 37 – does ‘structure’ = histology?

Line 58 – peculiarities? Probably should just delete this word

Line 130 – “testes were weighed individual after removal of the tunica?” The tunica does not need to be removed and probably would cause more artefacts than when testes are kept intact. Why was this approach used?

Line 163 – were the males also 45 days of age?

Line 170-172 – you could just state Spats1 KO or WT were mated with …

How long were they mated for?

Line 187 – is this a hybrid between EM and basic light histology approaches?

Line 228 – how were sperm recovered? By cutting the cauda or by retrograde perfusion?

Line 246 – typo: “autors”

Results:

Thank you to the authors for ageing the mice to 1 year to determine if there were any age-related phenotypes, this is a good approach.

Figure 3A – small suggestion – change the polka dot legend to show male first like in all the others

B – change the axis to g instead of gr – you could label these as Bi, Bii, Biii

D – these are quite beautiful and especially the lower right cross-section. It is clear that spermatogenesis looks normal, however the tubules on lower left and right and not stage matched. If you could change the WT to match the KO or vice versa that would be more appropriate. It is much simpler probably to just do normal 5 um sections and PAS stain the testes however for a similar result.

E- I do not think it was stated how concentration was measured but depending on the method used to extract sperm (if the caudae were simply nicked) then counting the resulting sperm is not accurate. The error bars are large so it might be useful to omit this panel. Truly accurate counts can only be done with homogenisation of the epididymis or the cauda specifically, as other methods do not fully recover the sperm from the epididymides.

F – progressive motility looks fine, however, the authors could also show the total motility and replace the panel to the left with this

Line 355 – no significance and it looks very similar for head defects, so it is best to just state no difference

Fig 4A – the legend could go underneath the columns

C – comment as above

B – The tubules are not stage-matched and are decent but not as good quality as in the above figure.

Discussion:

The discussion covers all topics of the m/s. Line 455 – a few hints – could be rephrased

6. PLOS authors have the option to publish the peer review history of their article (what does this mean?). If published, this will include your full peer review and any attached files.

Reviewer #1: No

Reviewer #2: No

---

## [Author Response · Author response to Decision Letter 0]

5 Apr 2021

Dear Editor,

 Thank you for considering our manuscript “SPATS1 (Spermatogenesis-associated, serine-rich 1) is not essential for spermatogenesis and fertility in mouse” ([PONE-D-21-03796] - [EMID:45ee38e7b541001a]) for publication in PLOS ONE. We thank all the comments and suggestions, that helped improve the ms. We are now submitting the corrected version. All the received comments from the reviewers have been considered, and we are hereby responding to all of them in a detailed fashion. Most of the suggested corrections have been included in the revised ms. 

Besides, following the journal requirements, we have provided the original uncropped and unadjusted Western blot image (S1_raw images), and amended the ethics statement to include the full name of the ethics committee that approved our specific study (now Lines 122-123 and 130-131 of the revised version with track changes).

Editor:

Please include additional information on the functional role of SPATS in relation to evolutionary process. The functional relevance of the genes analysed of a particular pathway be elaborated in the discussion. Reanalyze the sperm count with tissue homogenates and using stage specific sections for histological anlyses are recommended.

Reply: Thanks for the suggestions; they have been incorporated. 

Specifically, as indicated, SPATS1 is evolutionarily conserved along metazoans. Now, following the recommendations from Reviewer 1, we have included the alignment of SPATS1 from mouse with that of human, and also with the predicted SPATS1 sequences from several species, from invertebrates to mammals. As stated in the Discussion, reproduction-related proteins are in general poorly conserved (most probably due to unusually fast evolution rates), and therefore this high conservation would suggest an important, conserved role for SPATS1. However, despite its high evolutionary conservation, the functional role of SPATS1 is unknown. Moreover, as shown in this ms, it is not individually essential for male fertility.

Concerning the reanalysis of sperm count, we re-analyzed the sperm count in testicular cell suspensions by flow cytometry, which represents a widely accepted means to analyze testicular cellular content, with very high quantitative analytical power and statistical weight (now S2 Table). We also replaced the histological images with matched stages as suggested. A detailed point-by-point description is in the response to reviewers below.

Reviewer #1: 

1) In the abstract (i.e. last sentence) and throughout the text the authors should be careful about making parallels between Human fertility phenotypes related to SPATS1 and the mouse phenotypes described here. It is very possible – as the authors briefly note in the discussion – that one is not a good model for the other, as it is the case with many other spermatogenesis genes.

Reply: Done. Thank you for the comment. We have now clarified that SPATS1 is not essential for male fertility in mouse (see lines 45 and 95 of the revised version with track changes). Besides, although previously stated, we further emphasized that the studies that had suggested a link with male testicular health were from human and bull (line 539).

The gene is present, that seems clear. Yet, considering the scope of the study, I would recommend showing an alignment of mouse and human SPATS1 sequences (CDS and Proteins) to point to the regions that are indeed “conserved”. Please also carefully consider the next comments regarding SPATS1 evolution in the following points.

Reply: Done. Following the reviewer´s suggestion, in the revised version we have included an alignment of mouse and human SPATS1 sequences, both for proteins (newly included Fig 1B) and for CDS (newly included S2 Fig), to show the extensive similarities. We have also payed careful attention to the related comments addressed in the following points (see below).

2) Page 4 Line 78-80: What is the proposed function across evolution? Meiosis? Is it a sex-specific gene other metazoan?

Reply: There is no clue about the possible function. Yes, it is a testis-specific gene in every species for which we could find expression-related information in databases. The reviewer is right that “supports the idea that this protein could have an important, conserved role in testis” can be interpreted as if an important role for this protein had been proposed. As the idea is based on the high conservation level (in this regard, it must be recalled that reproductive proteins are in generally poorly conserved, as stated in the Discussion section), we have now replaced “supports the idea” with “allows to suspect” (line 79).

3) Page 12 Line 280-281: Clarify this sentence. What is the observation?

Reply: Done. The wording of the sentence has been changed to clarify the sentence. We hope it is clearer now (now lines 286-290).

4) The conclusion from figure 1 need to be developed to fully support the authors conclusions: the authors need to show an alignment or a few representative sequences to support the claim that it is “highly conserved” (again to distinguish between evolutionarily retained and conserved at the sequence level). Alternatively, they could use branch length on the tree to support this claim. Finally, the authors need to specify what type of tree is displayed and the scale used (substitutions per site maybe?) in Fig1.

Reply: Done. Thank you for the suggestion. We have now included an alignment of about 170 species (newly added S1 Fig). This alignment includes most of the species that were used to make the tree from Fig 1A. Besides, in the legend to Fig 1A we have explained that the tree is indeed a substitutions per site tree (it is represented to scale, with branch lengths representing the number of substitutions per site; lines 303-304), and the scale is shown in the figure. Sorry for not clarifying it before.

5) Page 13 line 299: What does differentially expressed means here? Relative to what? Please clarify.

Reply: Sorry, the reviewer is right. Spats1 is differentially expressed in the testis compared to other tissues. The sentence has been now reformulated to clarify this (now line 312).

6) Can the author specify how many backcrosses in which strain background was made to produce the experimental set of mice? 

Reply: Three backcrosses to C57BL/6J were done. This information has been now included in the text (lines 151-152).

7) Can the author specify where the antibody maps relative to the full-length protein and frame-shifted (truncated) sequence?

Reply: This was a commercial antibody, and the exact amino acidic sequence was proprietary information from the company. Anyway, by asking the company we could find out that it spanned the central region of the protein (as stated in the manuscript), around amino acid 186. As the frameshift in the KO protein is at aa 90 (thus missing the central and C-t regions, which are the most highly conserved along evolution) this antibody would not recognize the mutant protein. 

8) Page 17 Line 412-415: maybe the author could show results of these qPCRs for 4C cells.

Reply: Thank you for the suggestion. In fact, we are not sure whether the targets of the Wnt pathway should be mostly expressed in 4C cells. Although the highest SPATS1 expression levels were detected in spermatocytes (and even more so during the first spermatogenic wave), relatively high expression levels were also detected in other testicular cell types, including Sertoli cells, spermatogonia, and peritubular myoid cells. Furthermore, in embryonic testes SPATS1 is detected in gonocytes (at least in the rat). 

The targets of the Wnt pathway have been detected in different testicular cell types as well. Most reports about testicular disruption of the Wnt pathway show overexpression of Wnt targets in Sertoli cells and spermatogonia (e.g. in mutants with altered β-catenin expression). On one hand, aberrant activation of Wnt signaling targets in testis disrupts Sertoli cells differentiation and their ability to support spermatogenesis, resulting in seminiferous tubule degeneration and infertility. However, the different reports are contradictory concerning the developmental moment in which this effect is observed, ranging from embryonic life to adulthood (e.g. Boyer et al., 2008, doi 10.1095/biolreprod.108.068627; Chang et al., 2008, doi 10.1242/dev.018572; Tanwar et al., 2010, doi 10.1095/biolreprod.109.079335). On the other hand, increasing evidence suggests that both disruption and overexpression of the Wnt pathway specifically in male germline also affects spermatogenesis. For instance, a study in which the Wnt pathway was overactivated in the germ line through constitutive β-catenin expression, showed overexpression of the downstream targets (TCF1, Lef1, cyclin D1) in germ cells, and particularly in spermatogonia (Kumar et al., 2016, doi 10.18632/oncotarget.13920). Again, there are inconsistencies concerning the exact timing (e.g. Chang et al., 2011, doi:10.1371/journal.pone.0028039; Kerr et al., 2014, doi: 10.1095/biolreprod.112.105809; Kumar et al., 2016, doi 10.18632/oncotarget.13920; Chassot et al., 2017, doi: 10.1016/j.ydbio.2017.04.010). In addition, discrepancies between in vitro and in vivo results are not uncommon (e.g. Xue et al 2021, doi 10.1093/molehr/gaaa085).

As a consequence, due to the intricate signaling within the seminiferous epithelium (e.g. between somatic and Sertoli cells), and to the complexity of Wnt signaling itself, in case the Wnt pathway would be affected by the absence of SPATS1, we cannot be sure if we should expect the effect particularly in spermatocytes or in other cell types as well. This is the reason why we decided to analyze it in whole testes. 

This being said, we agree that it could be interesting to analyze the expression of Wnt targets in isolated cell types (i.e. 4C cell population, and maybe also 2C), to see if at least some subtle expression differences were detected in specific cell types. Unfortunately, due to Covid 19 pandemic, the transgenic facility significantly reduced its staff and workload, discontinuing most of the lines (including Spats1 KO) and freezing embryos. We must also take into consideration that in testicular studies in which some significant alterations of the expression levels of the Wnt pathway targets were observed, these were usually acccompanied by dramatic phenotypic changes (alterations in testis development, infertility, and/or testicular cancer [e.g Kerr et al., 2014; Kumar et al., 2016; Lanza et al., 2016, doi: 10.1080/15384101.2016; Chassot et al., 2017]). Therefore, as Spats1 KO mice do not present an overt phenotype, we think that the effort of reviving the mouse line to repeat the qRT-PCR assays in isolated cell populations is not worth it, as it is highly unlikely that even subtle changes in gene expression will be detected.

9) Page 17 Line 419: please indicate here what is the expected level and function of c-Myc in WT testes.

Reply: Done. We have now indicated it in the text of the revised version (now lines 439-440).

Briefly, the transcription factor/proto-oncogene c-Myc is one of the most common targets of the Wnt pathway that are activated by the β-catenin/TCF4 complex (He et al., 1998, doi 10.1126/science.281.5382.1509), together with Lef1, Tcf1, cyclin D1.

Particularly in WT testis, c-Myc has been shown to be expressed in Sertoli cells from 8-day-old rats but hardly detectable in cells from those aged 14 and 28 days, and a role in immature Sertoli cells has been suggested (Meroni et al 2019, doi: 10.3389/fendo.2019.00224). Interestingly, different reports have detected c-Myc overexpression in adult testes of mice and rats under abnormal conditions such as treatment with toxic substances (e.g. Wang et al., 2020, doi 10.1111/and.13841; Somade et al., 2020, doi: 10.1016/j.metop.2020.100051). 

In summary, c-Myc expression is not expected to be observed in WT adult testis. As the study referred in the manuscript (using heterologous cell culture transfections) suggested that SPATS1 would be a TCF4 repressor, we aimed at finding out whether c-Myc (as well as other common targets) was upregulated in Spats1 KO mice. Furthermore, since – as mentioned in the ms – a report suggested a possible link between Spats1 mutation and human testicular cancer, we found it interesting to analyze if the expression levels of c-Myc (a proto-oncogene) were altered in the testes of Spats1-deficient mice.

10) Page 21 Line 483: I would not be as categorical that SPATS1 is *not* a member of a multi-gene family. There is also SPATS2 and 2L in the mouse and human genome. Not to mention the SPATA gene family…

Reply: Thank you for the comment. To the best of our knowledge, SPATS2 and SPATS2L do not have any sequence similarity to SPATS1. BLAST alignments indicate that: “No significant similarity was found”, neither in mouse nor in human, and neither at the protein nor at the nucleotide levels (SPATS2 and 2L have partial similarity between them, though). Besides, according to our analyses, they do not share any common domain or other common trait. 

In the original paper where SPATS2 was described, and named p59scr (Senoo et al., Biochem Biophys Res Commun 292:992, 2002; doi: 10.1006/bbrc.2002.6769), the authors state: “The entire protein was rich in serine residues (13.6%), especially within amino acids between 373 and 411 (35.0%). Because of these characteristics, we named this novel protein p59scr (59 kDa, Serine-rich Spermatocytes and Round spermatid protein).” It is unfortunate that due to its overall serine content and testicular expression pattern, GenBank gave it a similar name to SPATS1, which may lead to confusion.

The ClustalW alignment for Mus musculus protein sequence between SPATS1 and SPATS2, below, shows the lack of similarity:

Spats1 ------------------------------------------------MESSKDTQHGDA

Spats2 MSRKQSQKDSSGFIFDLQSNTVLAQGGTFENMKEKINAVRAIVPNKSNNEIILVLQHFDN

 * ** * 

Spats1 LESKSCLANRTSSR------------------QNKRTSLSSSDGTGPRVTESLGLPRVLT

Spats2 CVDKTVQAFMEGSASEVLKEWIVTGKKKNKKKKSKPKPASEASGSAPDSSKSAPIQEEQP

 .*: * .* :.* .. *.:.*:.* ::* : . .

Spats1 PSDTAAELG---------------------------------------------------

Spats2 ASSEKGSINGYHVNGAINDAESVDSLSEGLETLSIDARELEDPEFAAAETLDRTGSVLEN

 .*. ..:. 

Spats1 ------------------QKTSSSSSSSSSSAQSNRSSKVSLPEIPK-------------

Spats2 GVSDFEPKSLTAHSISNVQQSRNAAKSLSRTTPGAQVSNLGMENVPLSSTNKKLGSNIEK

 *:: .::.* * :: . : *::.: ::* 

Spats1 ---------------------------EKYPEEFSLLNSQTEDGQRPEWTFYPRFSSNIH

Spats2 SVKDLQRCTVSLARYRVVVKEEMDASIKKMKQAFAELQSCLMDREVALLAEMDKVKAEAM

 :* : *: *:* * : . : :..:: 

Spats1 TYHIG------------------KQCFFNGVFRGNRRSVAERTVDNSLGKKKYDIDPRNG

Spats2 EILLSRQKKAELLKKMTDVAVRMSEEQLVELRADIKHFVSERKYDEDLGRVARFTCDVET

 :. .: : : . :: *:**. *:.**: : 

Spats1 IPKLTPGDNPYMFPEQSKEFFKAGATLPPVNFSLGPYEKKFDTFIPLEPLPKIP------

Spats2 LKQSIDSFGQVSHPKNSYSTRSRCSLVAPVSLSG-PSDGSAASSSPDASVPSLPGANKRN

 : : . . .*::* . . : :.**.:* * : . : * .:*.:* 

Spats1 NLPFWEKEKANN----------------------LKNEIKEVEELDN-------------

Spats2 CAPREASAAMTNSSDRPCQAHREVFPGNRRGGQGYRAQSQKTADPSNPGRHDSVGRYRNS

 * . .* : : ::. : .* 

Spats1 -WQVPMPFLHGFFSTGASNFSRQQ------------------------------------

Spats2 SWYSSGPRYQGVPPQAPGNAGERSRPYSAGTNGTGAISEPSPPKPSFKKGLPQRKPRASQ

 * . * :*. . ...* ..:. 

Spats1 ------

Spats2 AEAANS

 Regarding the Spata (spermatogenesis-associated) family, despite that one of the synonymous names that appear in GenBank for human (but not for mouse) SPATS1 is SPATA 8, Spats1 is unrelated to the rest of Spata genes. Moreover, a tblastn search among all GenBank conducted with mouse SPATS1 sequence and excluding Spats1 (in order to find out if there were possible matches with other sequences but Spats1 homologous sequences) gives the message “Not significant similarity was found”, not producing any matches with SPATS2, the products of any Spata gene, nor any other predicted proteins in GenBank database.

Reviewer #2: 

Line 32 – SPATS1 loss of function or overexpression? – I note this is further fleshed out well below in lines 81-91

Reply: Done. We have now clarified this in the text (line 33).

Line 37 – does ‘structure’ = histology?

Reply: Thanks for the observation. It has been amended by replacing “structure” by “histology” (now line 38).

Line 58 – peculiarities? Probably should just delete this word

Reply: Done. It has been deleted.

Line 130 – “testes were weighed individual after removal of the tunica?” The tunica does not need to be removed and probably would cause more artefacts than when testes are kept intact. Why was this approach used? 

Reply: The reviewer is right that removal of the tunica albuginea may cause some distortion. However, we are used to removing the tunica (which we do very cleanly), because we usually employ the tissue for preparing testicular cell suspensions. Besides, we wanted to get an idea of the weight of the testicular parenchyma. 

Line 163 – were the males also 45 days of age?

Reply: Yes, they were also 45 days of age. It has been now clarified (now line 167).

Line 170-172 – you could just state Spats1 KO or WT were mated with …

How long were they mated for?

Reply: Done. The sentence was reformulated according to the reviewer´s suggestion, and the mating time (three months) was included (now lines 174-175). 

Line 187 – is this a hybrid between EM and basic light histology approaches? 

Reply: We usually prefer this method for sample preparation for microscopy because it involves a double fixation, first with glutaraldehyde and then with osmium tetroxide, thus enabling an excellent preservation of testicular histology. Besides, although in this case semi-thin sections and optical microscopy were used, the same samples can be eventually employed for thin sections in case transmission electron microscopy (TEM) would be required. 

Line 228 – how were sperm recovered? By cutting the cauda or by retrograde perfusion? 

Reply: Sperm were recovered by cutting the cauda. We have now explained it in the text (now line 234). 

Line 246 – typo: “autors”

Reply: Sorry, thank you for noting it. We have corrected it (now line 252).

Figure 3A – small suggestion – change the polka dot legend to show male first like in all the others

Reply: Done, thank you for the suggestion.

B – change the axis to g instead of gr – you could label these as Bi, Bii, Biii

Reply: Done. gr has been replaced by g, and the different parts have been labeled as suggested.

D – these are quite beautiful and especially the lower right cross-section. It is clear that spermatogenesis looks normal, however the tubules on lower left and right and not stage matched. If you could change the WT to match the KO or vice versa that would be more appropriate. It is much simpler probably to just do normal 5 um sections and PAS stain the testes however for a similar result.

Reply: Done. Thanks for the suggestion. We have replaced the lower left image (WT), to match stage with the lower right (KO). 

E- I do not think it was stated how concentration was measured but depending on the method used to extract sperm (if the caudae were simply nicked) then counting the resulting sperm is not accurate. The error bars are large so it might be useful to omit this panel. Truly accurate counts can only be done with homogenisation of the epididymis or the cauda specifically, as other methods do not fully recover the sperm from the epididymides. 

Reply: A fixed number of nicks were performed in each cauda (it is now explained in Materials and Methods, line 234). Following the reviewer´s recommendation, in the revised version we have omitted this panel from Fig 3E, and from Fig 4C as well.

We agree that this method is not accurate as it does not fully recover the sperm. However, we have made exactly the same cuts in each cauda, and let the sperm swim out for exactly the same time lapse (15 min) in the same conditions and in exactly the same volume, and it was performed by the same operator. Therefore, we think that - although not accurate in absolute numbers - the data is comparable between samples. As it can be useful to show the absence of significant differences between WT and KO (the error bars are large because there were individual differences, but not between WT and KO), we included it now as a supplementary figure (S3 Fig). However, we can still remove it if the reviewer disagrees. 

Besides, we re-analyzed the sperm count in testicular cell suspensions by flow cytometry, which, as stated above, represents a widely accepted means to analyze testicular cellular content, with very high quantitative analytical power and statistical weight (now S2 Table). 

F – progressive motility looks fine, however, the authors could also show the total motility and replace the panel to the left with this

Reply: Done. We have replaced Fig 3E with a panel showing total motility.

Line 355 – no significance and it looks very similar for head defects, so it is best to just state no difference

Reply: Thanks for the suggestion. We have modified it according to the reviewer´s suggestion, indicating that no significant differences were found (now lines 368-370).

Fig 4A – the legend could go underneath the columns

Reply: Thank you for the suggestion. We tried placing the legend underneath the columns, but we got the impression that it did not look good, as it was too long (because it indicates a cross). Besides, the style is standardized with that in Fig 3A that also shows the results of fertility tests.

C – comment as above

Reply: Done. Following the reviewer´s recommendation, we have eliminated Fig 4C, and replaced it with a panel showing total motility.

B – The tubules are not stage-matched and are decent but not as good quality as in the above figure.

Reply: Done. Thank you for the suggestion. We have changed the images and replaced them with stage-matched tubules.

Discussion:

The discussion covers all topics of the m/s. Line 455 – a few hints – could be rephrased

Reply: Done. We have rephrased the sentence, by replacing “A few hints” with “Some data have allowed us to suspect…” (now line 477).

 We hope that with these corrections and clarifications, the ms will now be suitable for publication in PLOS ONE. In case there would be any additional corrections or suggestions, please let us know.

Sincerely,

Adriana Geisinger

Corresponding autor

---

## [Decision Letter · Decision Letter 1]

19 Apr 2021

SPATS1 (Spermatogenesis-associated, serine-rich 1) is not essential for spermatogenesis and fertility in mouse

PONE-D-21-03796R1

Dear Dr. Geisinger,

We’re pleased to inform you that your manuscript has been judged scientifically suitable for publication and will be formally accepted for publication once it meets all outstanding technical requirements.

Kind regards,

Suresh Yenugu

Academic Editor

PLOS ONE

Additional Editor Comments (optional):

Reviewers' comments:

Reviewer's Responses to Questions

**Comments to the Author**

1. If the authors have adequately addressed your comments raised in a previous round of review and you feel that this manuscript is now acceptable for publication, you may indicate that here to bypass the “Comments to the Author” section, enter your conflict of interest statement in the “Confidential to Editor” section, and submit your "Accept" recommendation.

Reviewer #1: All comments have been addressed

Reviewer #2: All comments have been addressed

2. Is the manuscript technically sound, and do the data support the conclusions?

Reviewer #1: Yes

Reviewer #2: Yes

3. Has the statistical analysis been performed appropriately and rigorously? 

Reviewer #1: Yes

Reviewer #2: Yes

4. Have the authors made all data underlying the findings in their manuscript fully available?

Reviewer #1: Yes

Reviewer #2: Yes

5. Is the manuscript presented in an intelligible fashion and written in standard English?

Reviewer #1: Yes

Reviewer #2: Yes

6. Review Comments to the Author

Reviewer #1: In their responses to the reviewers, the authors have carefully addressed all of the comments.

I foresee no further revisions at this point.

Reviewer #2: Thank you for your response to all my questions and comments. I think the manuscript now stands in an improved state and is ready for publication.

7. PLOS authors have the option to publish the peer review history of their article (what does this mean?). If published, this will include your full peer review and any attached files.

Reviewer #1: No

Reviewer #2: No

---

## [Editor Report · Acceptance letter]

21 Apr 2021

PONE-D-21-03796R1 

SPATS1 (Spermatogenesis-associated, serine-rich 1)is not essential for spermatogenesis and fertility in mouse 

Dear Dr. Geisinger:

I'm pleased to inform you that your manuscript has been deemed suitable for publication in PLOS ONE. Congratulations! Your manuscript is now with our production department. 

Kind regards, 

on behalf of

Dr. Suresh Yenugu 

Academic Editor

PLOS ONE